# Antimicrobial Peptide against *Mycobacterium Tuberculosis* That Activates Autophagy Is an Effective Treatment for Tuberculosis

**DOI:** 10.3390/pharmaceutics12111071

**Published:** 2020-11-09

**Authors:** Erika A. Peláez Coyotl, Jacqueline Barrios Palacios, Gabriel Muciño, Daniel Moreno-Blas, Miguel Costas, Teresa Montiel Montes, Christian Diener, Salvador Uribe-Carvajal, Lourdes Massieu, Susana Castro-Obregón, Octavio Ramos Espinosa, Dulce Mata Espinosa, Jorge Barrios-Payan, Juan Carlos León Contreras, Gerardo Corzo, Rogelio Hernández-Pando, Gabriel Del Rio

**Affiliations:** 1Department of Biochemistry and Structural Biology, Institute of Cellular Physiology, National Autonomous University of Mexico (UNAM), 04510 Mexico City, Mexico; erika16pc@hotmail.com (E.A.P.C.); Christian.Diener@systemsbiology.org (C.D.); 2Experimental Pathology Section, National Institute of Medical Sciences and Nutrition Salvador Zubirán, 14080 Mexico City, Mexico; panterarosa860919@hotmail.com (J.B.P.); octavioramos13@gmail.com (O.R.E.); dulmat@yahoo.com.mx (D.M.E.); qcjbp77@yahoo.com.mx (J.B.-P.); jcleonc@hotmail.com (J.C.L.C.); 3Department of Neurodevelopment and Physiology, Instituto de Fisiologia Celular, National Autonomous University of Mexico, 04510 Mexico City, Mexico; gabrielm@ifc.unam.mx (G.M.); dmoreno@ifc.unam.mx (D.M.-B.); scastro@ifc.unam.mx (S.C.-O.); 4Laboratorio de Biofisicoquímica, Departamento de Fisicoquímica, Facultad de Química, Universidad Nacional Autonoma de Mexico, 04510 Mexico City, Mexico; costasmi@unam.mx; 5Department of Molecular Neuropathology, Institute of Cellular Physiology, National Autonomous University of Mexico (UNAM), 04510 Mexico City, Mexico; tmontiel@ifc.unam.mx (T.M.M.); lmassieu@ifc.unam.mx (L.M.); 6Department of Molecular Genetics, Institute of Cellular Physiology, National Autonomous University of Mexico, 04510 Mexico City, Mexico; suribe@ifc.unam.mx; 7Department of Molecular Medicine and Bioprocesses, Institute of Biotechnology, National Autonomous University of Mexico (UNAM), 62210 Cuernavaca Morelos, Mexico; corzo@ibt.unam.mx

**Keywords:** antimicrobial peptide, tuberculosis, autophagy, iztli peptide, multidrug resistant

## Abstract

*Mycobacterium tuberculosis* (MTB) is the principal cause of human tuberculosis (TB), which is a serious health problem worldwide. The development of innovative therapeutic modalities to treat TB is mainly due to the emergence of multi drug resistant (MDR) TB. Autophagy is a cell-host defense process. Previous studies have reported that autophagy-activating agents eliminate intracellular MDR MTB. Thus, combining a direct antibiotic activity against circulating bacteria with autophagy activation to eliminate bacteria residing inside cells could treat MDR TB. We show that the synthetic peptide, IP-1 (KFLNRFWHWLQLKPGQPMY), induced autophagy in HEK293T cells and macrophages at a low dose (10 μM), while increasing the dose (50 μM) induced cell death; IP-1 induced the secretion of TNFα in macrophages and killed Mtb at a dose where macrophages are not killed by IP-1. Moreover, IP-1 showed significant therapeutic activity in a mice model of progressive pulmonary TB. In terms of the mechanism of action, IP-1 sequesters ATP in vitro and inside living cells. Thus, IP-1 is the first antimicrobial peptide that eliminates MDR MTB infection by combining four activities: reducing ATP levels, bactericidal activity, autophagy activation, and TNFα secretion.

## 1. Introduction

*Mycobacterium tuberculosis* (MTB) is a bacterium that has accompanied humans since their migration from Africa [1]; it has been responsible for tuberculosis (TB) that caused the death of millions of people in the past [2] and currently represents a major risk to health worldwide [3]. Consistent with its history, MTB has evolved many mechanisms to deal with different antibiotic compounds used to treat MTB infections. In fact, MTB is an intracellular bacterium and much of its infection cycle is developed within its principal host cell, the macrophage. Indeed, macrophages contribute to the elimination of MTB via diverse mechanisms, such as the production of reactive products of oxygen and nitrogen, as well as the successful acidification and maturation of phagosomes; however, as a consequence of the chronic, enduring nature of this infection, MTB has evolved to detect, react to, and manipulate these intracellular mechanisms to avoid their destruction, such as delaying the phagosome–lysosome fusion to create a suitable environment for bacillary survival and replication (reviewed in [4]). At the same time, macrophages have evolved alternative mechanisms to eliminate MTB, such as the induction of autophagy that facilitates phagosome–lysosome fusion and the bacilli clearance [5]. In the present work, we characterize a synthetic peptide that, in addition to its antibiotic activity against MTB, activates a natural response of the cell to deal with MTB infection, namely autophagy.

Autophagy is a catabolic process that maintains cellular homeostasis by degrading different cellular components [6]. Macroautophagy (hereby referred to as autophagy) is a specific type of autophagy that eliminates damaged organelles or intracellular bacteria (xenophagy). The cargo to be degraded is first recognized and engulfed by a specialized vesicle named autophagosome, which forms de novo from an isolated membrane or phagophore. The autophagosomes fuse with lysosomes to form autolysosomes, and the engulfed cargo is degraded by lysosomal hydrolases. Autophagy activation, cargo selection, and autophagosomes maturation are regulated by an evolutionarily conserved molecular machinery (reviewed in [7]). Relevant to mention in this work are LC3-II and p62/SQSTM1. LC3-II is required for phagophore elongation around the cargo, forming the autophagosome. LC3-II also attracts cargo receptors that contain both an LC3-interacting motif and a specific binding motif for a label in the cargo, such as ubiquitin in the case of p62/SQSTM1. Since cargo receptors are degraded together with the engulfed cytoplasmic material, a reduction in the amount of p62/SQSTM1 is indicative of a functional autophagic flux [8].

The induction of autophagy is known to help cells address different pathological states, such as cancer [9], neurodegeneration [10], or intracellular infections [11]. In the latter case, it has been proposed that antimicrobial peptides (AMPs) capable of inducing autophagy may represent a novel mechanism to deal with pathogens resistant to commonly used antibiotics and autophagy, such as MTB infection [12,13]. AMPs display multiple functions such penetrating cells [14] or modulating the host immune response [15]. In this regard, the relationship between infectious agents and autophagy was documented back in 1984 [16] and more recently it has been reported that autophagy produces AMPs by proteolysis of cytosolic proteins of infected cells [17,18]. In fact, three AMPs (Indolicidin and two peptides derived from Seminal plasmin) were shown to induce autophagy in Leishmania cells [19]. Hence, this primary evidence suggests that in response to infection, cells may activate autophagy to produce AMPs to kill the infecting bacteria either through AMP direct antibiotic activity or through autophagy clearance of bacteria. Direct evidence for the effectiveness of AMPs to activate autophagy for bacterial clearance was reported by the Agerberth group showing that Vitamin D induces the expression of an AMP (LL-37) that in turn induces autophagy that eliminates MTB from infected cells [20]. It is relevant to consider that LL-37 kills bacteria in the microMolar range (2–10 μg/mL or 0.4–2 μM [21]), hence, at that concentration it also induces autophagy supporting the notion that these combined activities represent a natural strategy to combat infectious diseases.

Thus, to aid the immune system to cope with resistant infections, it would be desirable to have access to AMPs that at a given concentration kill bacteria and induce autophagy in the host cells without significant toxicity on healthy cells. How may AMPs achieve this dual role, to induce autophagy and kill bacteria, at the appropriate dosage? One possible mechanism may involve mitochondria function, since mitochondria share membrane similarities with bacteria [22]; thus, mitochondrial damage might be a signal for cells to induce mitophagy, the autophagic process activated by cells to selectively eliminate dysfunctional mitochondria [23]. In agreement with this idea, a recent report has shown that two receptors involved in the xenophagic elimination of intracellular infections also participate in the targeting and degradation of mitochondria [24]. This is in agreement with the protective role of mitophagy during bacterial infections, helping to maintain the production of reactive-oxygen species and other immune responses associated with mitochondrial function [25]. Such a mechanism may be able to deal with the inhibition of phagosome maturation evolved by MTB [26].

We have previously reported that a synthetic AMP, Iztli peptide 1 (IP-1: KFLNRFWHWLQLKPGQPMY), penetrates mammalian cells and make pores in membranes with high electric potential values, such as mitochondrial or bacteria membranes [27]. Cell-penetrating peptides are known to induce autophagy [28] hence IP-1 can combine the antimicrobial and induction activities of autophagy, making it a candidate to treat resistant infections. In the present work we first show that IP-1 is capable of inducing autophagy in mammalian cells, HEK293T cells, and macrophages, at concentrations where no toxicity in these cells is observed, and show that its mechanism of action was not mediated by altering mitochondrial integrity, but one of its products. In fact, we found that IP-1 binds to ATP in vitro and it also decreased intracellular ATP concentration. In the second part of this work, we show that infected MTB macrophages incubated with IP-1, at the same concentrations IP-1 killed MTB in vitro, showed significant bacterial clearance concomitant with more autophagosomes and with a high production of the crucial protective cytokine TNFα. Moreover, a significant efficient therapeutic activity was observed in vivo, in a model of progressive pulmonary TB induced by drug sensitive or resistant strains in BALB/c mice. We propose that the ability of IP-1 to sequester ATP decreases its intracellular concentration inducing autophagy, which helps to eliminate MTB, in addition to the antimicrobial activity of the IP-1 peptide.

## 2. Materials and Methods

### 2.1. Cell Culture and Reagents

Primary Mouse Embryonic Fibroblasts (MEFs) were prepared from CD-1 mice embryos of 17–18 days of gestation following standard procedure. The mice were obtained from the animal house at the Instituto de Fisiología Celular, UNAM, following IACUC guidelines (approved protocol CICUAL SCO51-18). Cells were grown in DMEM (10,569, Gibco, Grand Island, NY, USA) supplemented with 10% fetal bovine serum (16,000, Gibco) and penicillin/streptomycin 100 U/mL (15,140, Gibco) in the presence of 5% of CO_2_ at 37 °C. The cells were incubated for 6 h with IP-1 peptide at different concentrations (0, 20, 30, 40 and 50 μM) prepared in DMEM. HEK2913T cells were seeded in 12-well plates (1 × 105 HEK293T cells per well) in 1 mL of supplemented DMEM (Invitrogen, Grand Island, NY, USA).

Iztli peptide 1 (IP-1, KFLNRFWHWLQLKPGQPMY) was synthesized and purified by Anaspec, Inc. (Fremont CA 94555, USA). The disaccharide D-(+)-Trehalose dihydrate (T0167-100G SIGMA-ALDRICH, St. Louis, MO, USA) was used as a positive control of autophagy induction at 100 mM for 5 h. Chloroquine was used at a final concentration of 40 µM for thirty minutes before treatment with IP-1 or trehalose. All reagents were diluted in water and used at indicated concentration in each experiment.

### 2.2. Autophagosomes and Autolysosomes Induction

Autophagosomes were detected using Cyto-ID kit (Enzo Life Sciences, ENZ-51031-K200, Farmingdale, NY, USA) following manufacturer´s recommendation. Briefly, HEK293T cells or MEFs were seeded in 12-well plates (1 × 10^5^ HEK293T cells or 3 × 10^4^ MEF cells per well) 18 h before treatment. Nuclei were stained with Hoechst 33,342 (excitation at 350 nm) whereas autophagosomes were stained with Cyto-ID kit (excitation at 488 nm). The samples were analyzed in an inverted epifluorescence microscope (Nikon ECLIPSE Ti-U) coupled to a camera and photographs were directly taken from the well plates. In each experiment at least two photographs for every condition were taken and at least three independent experiments were performed. The photographs were then analyzed using Fiji [29]. Each photograph captured several cells, so each single cell was selected based on the nucleus staining as a single Region Of Interest (ROI) and for every ROI, particles (autophagosomes) in the green channel were counted using the algorithms included in Fiji for such goal. The cells were counted as having active autophagy if a cell contained ≥6 particles; in brief, background noise was first removed from the image using the “rolling Ball background subtraction” Fiji method. This was followed by manual thresholding in the fluorescence channel until only the foci were visible. Finally, particles were filled in using the “Watershed” function in Fiji and counted using the “Analyze Particles” step. The distribution of this data did not follow a normal distribution, hence Fisher’s exact test was used to assess the *p*-values with a significance threshold of 0.05. To determine the localization of IP-1 inside autophagosomes, a fluorescent peptide was used (TAMRA-IP-1) in combination with CytoID staining. 120,000 macrophage cells (J774) were seeded onto sterile Lab-Tek II chamber slides with cover (Nalgene Nunc, Rochester, NY, USA). The cells were incubated in CytoID (1 mL CytoID/mL cell culture medium) for 30 min at 37 °C, 5% CO_2_ and fixed prior to analysis. The cells were imaged at 63X using a confocal microscope Zeiss LSM 800 and processed using Fiji software. Autophagosomes were also identified by expressing both EGGF-LC3 and pmRFP-LC3 and observed by confocal microscopy as previously described [30]. EGFP is sensitive while pmRFP-LC3 is resistant to low pH. Therefore, the observation of both EGFP-LC3 and pmRGP-LC3 in the same vesicle indicates it is an autophagosome, as they have a neutral pH. Upon fusion with lysosomes, mature autolysosomes acquire an acidic pH and thus the GFP signal is lost. Time-lapse recording in a confocal Microscope Leica TSC- SP 5-11 was used to follow autolysosome maturation. Multiple photos of slices of the same field (z-stacks) were taken every 20 min for 2 h. For transfection, 10^6^ HEK293T cells were seeded on 35 mm dishes (Corning, NY, USA) for 16 h and then both pmRFP-LC3 and EGFP-LC3 expression vectors were co-transfected with PEI. pmRFP-LC3 was kindly provided by Tamotsu Yoshimori (Addgene plasmid # 21075) and EGFP-LC3 was kindly provided by Karla Kirkegaard (Addgene plasmind # 11546). Three to five independent experiments were performed, as indicated in each figure.

### 2.3. Western Blots

For Western blot assays of LC3, 5 × 10^5^ HEK293T cells were seeded in 6-well plates (Costar, USA) in 2 mL of supplemented DMEM (Invitrogen, Grand Island, NY, USA). After treatment with 10 μM of IP-1 or control without peptide, the cells were collected and lysated with a lysis buffer (Tris 50 mM, NaCl 150 mM, 1% NP-40, 0.5% Sodium deoxicolate, 0.1% SDS and complete ULTRA protease inhibitor cocktail (ROCHE, Germany). The amount of total proteins in each sample was estimated by measuring the absorbance at 280 nm using a nanodrop (ThermoFisher, Waltham, MA, USA) following vendor’s instructions. For each sample, 5 μg of total protein was loaded in a 15% polyacrylamide gel. After protein separation by SDS-PAGE, the gels were transferred to a PVDF membrane and incubated with primary antibodies, LC3B antibody (Cell Signaling, Beverly, MA, USA, cat. #2775), p62/SQSTM1 (Cell Signaling cat. #5114, USA) were diluted 1:1000 in TBST buffer containing 5% Albumin (Rocky Mountain Biologicals, Missoula, MT, USA) and incubated overnight at 4 °C. The next day, the membranes were washed with TBST and incubated for one hour with rabbit secondary antibodies HRP-conjugated Anti-Rabbit IgG Concentrate (SIGMA-ALDRICH, St. Louis, MO, USA, RABHRP1) 1:4000 in TBST with 3% Nonfat dry milk (Blotting-Grade Blocker, BIO-RAD Laboratories, Inc., Hercules, CA, USA, Cat.# 170-6404). The membranes were developed by chemioluminiscence (Immobilon Western Chemiluminescent HRP Substrate, Millipore Corporation, Billerica, MA 01821, USA, Cat. WBKLS0100). Digitalized blots were analyzed with the Fiji software (Version 2017) [29].

### 2.4. Cytotoxicity Assay

Cell viability after treatment with IP-1 was evaluated by trypan blue exclusion. Briefly, the cells were seeded (5 × 10^4^ for HEK293T and 3 × 10^4^ for MEF cells) in 12-well plates and treated for 6 h with 20 μM, 30 μM, 40 μM, and 50 μM of IP-1, collected and stained with trypan blue solution at 0.04% (Trypan Blue Stain (0.4%) Gibco, Life Technologies, Carlsbad, CA, USA). Alive and dead cells were counted to determine viability. Triplicates of three independent experiments were performed (9 samples total).

MEFs cells were incubated for 6 h with IP-1 peptide at different concentrations (0, 20, 30, 40, and 50 μM) prepared in DMEM. HEK2913T cells were seeded in 12-well plates (1 × 10^5^ HEK293T cells per well) in 1 mL of supplemented DMEM (Invitrogen, USA) and exposed to the same IP-1 concentrations as MEF cells.

In the case of macrophages (J774.1), the cytotoxicity was inferred from a colorimetric assay [31]. Briefly, 15,000 cells per well were grown in a 96-well plate flat-bottom at 37 °C in RPMI 1640 media (Invitrogen Life Technologies, Carlsbad, CA, USA) with 10% Serum Fetal Bovine (Gibco, USA) and 5% of CO_2_ for 24 h, then IP-1 was added to the cells while some other wells were not added any peptide, instead, DMSO or excipient solution of IP-1 was added (water).The cells were cultivated for 48 h and then fixated with glutaraldehyde 1% for 10 min and then were rinsed with PBS to finally let them dry at room temperature. Then, 80 μL of solution of crystal violet 0.1% (dissolved in 200 mM formic acid buffer at pH 6) was added and shake gently for 10 min; afterwards the cells were washed 3 times with de-ionized water, the plates were air-dried and finally add 200 μL of acetic acid solution at 10% in water for 10 min and shaken gently to solubilize and extract the violet stain. Then, 100 μL of each well was transferred to another 96-well plate with flat bottom and read the optical density at 570 nm in a spectrophotometer EPOCH2 (BioTek, Winooski, VT, USA), the larger the number of live cells. Three independent experiments were performed.

### 2.5. TUNEL Assay

As a positive control for TUNEL in MEFs, 2 μM Staurosporine was incubated for 2 h; for HEK293T cells, 400 μM Etoposide was incubated for 2 h. TUNEL assay was done according to manufacturer´s instructions (In Situ Cell Death Detection Kit, TMR red, ROCHE^®^ Cat. No. 12 156 792 910, Basel, Switzerland). Briefly, the cells were washed with PBS, fixed with 4% paraformaldehyde in PBS, washed again with PBS, and incubated with permeabilization solution (0.5% Triton X-100) for 5 min. Then, the cells were washed with PBS and incubated with TUNEL reaction in a humidified atmosphere for 60 min at 37 °C in the dark. The cells were washed three times with PBS. Images of cells were obtained by fluorescence microscopy (Nikon ECLIPSE Ti-U) and analyzed with the NIS Elements, Basic Research software, Version 3.13.

### 2.6. Annexin V Assay

The protocol was done following the manufacturer’s instructions (ApoAlert^®^ Anexin V from Clontech Cat. No 630110, Montain View, CA, USA). After treatment, the cells were washed with 1X Binding buffer^®^, and then 200 μL of staining mix (5 μL of Annexin V and 10 μL of propidium iodide in Binding Buffer) were added and incubated at room temperature for 15 min in the dark. Then, the cells were fixed with 4% paraformaldehyde in PBS. Post-fixation step, cells were washed with PBS. Finally, coverslips with the cells were placed on slides using mounting medium (VECTASHIELD^®^). The slides were analyzed under fluorescence microscope (Nikon Eclipse Ti-U). Images were processed using NIS Elements, Basic Research (NIKON INSTRUMENTS Inc^®^, Melville, NY, USA) software, Version 3.13.

### 2.7. Mitochondria Function

#### 2.7.1. Oxygen Uptake

Cell respiration was measured using the high-resolution oxygraph Oroboros with 2 mL chambers volume. The chambers were preheated and maintained at 37 °C. Basal respiration was measured in DMEM medium. After assessing steady respiration in basal state, we tested different inhibitors or promoter of oxygen uptake and compared them with the activity of the IP-1. FCCP (carbonyl cyanide-ptryfluoromethoxyphenylhydrazone), a decoupler of the oxidative phosphorylation, was used as a positive control of respiration at a concentration of 1 μM. Cyanide, a known inhibitor of electron transport in the mitochondrial respiratory chain, was used as negative control of respiration at a concentration of 100 mM. Iztli peptide 1 was used at concentrations of 10 and 50 μM. The statistical analysis was performed using python (2.7, Python software foundation, Fredericksburg, VA, USA).

#### 2.7.2. Mitochondrial Membrane Potential In Situ

The in situ membrane potential of HEK293T cells was measured using tetraethylbenzimidazolylcarbocyanine iodide (JC-1), a cationic dye that has affinity for negatively charged membranes and, hence, accumulates in active mitochondria. The cells were incubated for 18 h in DMEM media with constant CO_2_ at 5%. After that time, the cells were detached and washed with PBS and incubated for ten minutes at 37 °C with JC-1 in DMEM. The cells were washed again and incubated with IP-1 diluted in DMEM medium at concentrations of 10 μM and 50 μM for either 15 min or an hour at 37 °C. After the treatment, the cells were rinsed with PBS and re-suspended again for fluorescence measure using the AMNIS imaging flow cytometer coupled to a multiphotonic microscope. The collection and analysis of data was performed with the proprietary software provided with the imaging cytometer.

### 2.8. Lipid Profile of Cells Treated with IP-1

#### 2.8.1. Infrared Detection of Lipids in Living Cells

FTIR spectra of HEK293T cells treated with Rapamycin (200 nM), IP-1 (10 μM), Substance P (100 nM) or none for the control were acquired in liquid environment using CaF2 modified devices with a constant height of 8 µm as described in Birarda [32]. The spectra were acquired at SISSI beamline at Elettra Sincrotrone Trieste on a Vis-IR microscope Hyperion 3000 coupled to a Vertex 70 interferometer (Bruker Optics GmbH, Ettlingen, Germany). Microscope knife-edge apertures were fixed at 30 µm × 30 µm gathering the signal of 2–4 cells. An MCT detector with a 100 µm sensitive element was used. The spectra were obtained from averaging 256 scans in the 6000–8000 cm^−1^ region in transmission mode using a 15× condenser/objective at a spectral resolution of 4 cm^−1^. Blackman-Harris 3-term apodization function and a filling factor of 2 were used. To each measurement a buffer point was also measured in a close position and both spectra were rationed against air background. Data were preprocessed by vector normalization spectra and were subtracted from a buffer spectrum obtained near to the sample. Hierarchical cluster analysis was performed on the 3000–2800 cm^−1^ infrared region, using the Ward algorithm. Principal component analysis was performed (Second derivative of subtracted spectra (9 smoothing points, Savitsky–Golay algorithm)) on the same region. The first two principal components represent 75.9% of the variance in the data. The analysis was performed in Python, using numpy and matplotlib.

#### 2.8.2. Mass Spec for NEFA Measurements

Mass spectrometry on HEK293T cells was used to determine the non-sterified free fatty acids (NEFAs). They were cultured in 10 cm petri dishes with DMEM medium supplemented with 10% Fetal Bovine Serum, Penicilin/Streptomycin at 37 °C in a 5% CO_2_ atmosphere. After the cells reached 70–90% confluency, they were treated with water, IP-1 10 µM, or IP-1 50 µM. Then, the cells were frozen and taken to the Lipidomics core in UCSD to obtain the concentration of each NEFA. Briefly, 500 μL of methanol was added to 500 μL of cell homogenate and 25 μL of 1 N HCl; 100 μL of 0.1 ng of internal control. The mixture was vortexed, and 1 mL of Iso-octane was added, vortexed, and centrifuged at 3000 rpm for 1 min to extract the top layer and dried it in speed vacuum. After this, 50 μL were taken for GC-MS (Agilent 5975 gas chromatograph with Mass Selective Detector, Santa Clara, CA, USA) analysis.

### 2.9. ATP Quenching Activity of IP-1

#### 2.9.1. Luciferase Activity in the Presence of IP-1

IP-1 binding to ATP was inferred by using the EnzyLight ATP Assay Kit (BioAssay Systems, Hayward, CA, USA). Equimolar concentrations of ATP and IP-1 (10 µM) were incubated in water for one hour and then ATP concentration was measured according to the luminiscence emitted by samples containing or not IP-1. The luminiscence was measured with a multi-mode microplate reader SynergyMx (Biotek Instruments, VT, USA).

#### 2.9.2. Isothermal Titration Calorimetry

The binding essay was performed at 25 C using a VP-ITC microcalorimeter (Microcal, Malvern, Worcestershire, UK). Aqueous 10 mM ATP and 0.25 mM IP-1 solutions were placed in the syringe and cell, respectively. The dilution experiment (ATP in syringe at 10 mM titrated over pure water) was also performed. The titrations consisted of first a blank 1 µL injection, followed by 28 injections of 10 µL, with a 13 min delay between them and stirring the cell content at 440 rpm. The raw heatflow data were processed using the ITC data analysis program provided by MicroCal. After subtraction of the dilution contribution, the thermodynamic parameters (equilibrium constant and binding enthalpy) for the ATP/IP-1 interaction were obtained using a 1:1 stoichiometry model, which satisfactorily fit the heat data. The affinity constant was found to be 440 µM with a binding enthalpy of −2100 cal/mol.

### 2.10. Intra and Extra Cellular Quantification of ATP

ATP levels were determined 15, 30, and 60 min after the exposure of HEK293T cells to 10 and 50 μM of IP-1. ATP determinations were performed using previously described methodology [33]. Briefly, the extracellular medium of HEK cells was collected, the cells were washed twice with pre-warmed Locke’s solution containing (in mM): 154 NaCl, 5.6 KCl, 3.6 NaHCO_3_, 2.3 CaCl_2_, 5 HEPES, 5.6 glucose, pH 7.4, and lysed by incubation in 60 μL somatic cell ATP releasing agent (Sigma FL-SAR). 10 μL of lysates or 100 μL of extracellular medium were used to measure ATP levels by means of a luminometer (Bio Orbit) using the luceferin-luciferase Chemiluminiscent kit (Molecular Probes A22066). The luminometer records quimioluminescence values in millivolts and ATP concentrations were calculated from readings obtained from an ATP standard curve (1.95–250 pmol). Aliquots of cell homogenates were kept for protein determination by the micro-Lowry´s method and data are expressed as pmol/μg of protein.

### 2.11. Image Analysis and Statistics

The Fiji platform was used to perform image analysis of both microscopy and western blot images [29], unless otherwise stated. Plots and statistical analysis were performed using the R programming language (http://r-project.org). The data and scripts used are available [34]. Statistical analysis was performed using one-way analysis of variance (ANOVA) followed by the Fisher´s post hoc test for multiple comparisons, unless stated otherwise.

### 2.12. Mycobacterium Tuberculosis Inhibition Assay

For this assay, the IP-1 peptide was tested by broth microdilution inhibition assay using the virulent strain of *M. tuberculosis* H37Rv (ATCC 27294) and a multi-drug resistant (MDR) strain CIBIN99, which is a clinical isolate resistant to all the primary antibiotics [35]. Drug susceptibility testing of *M. tuberculosis* was performed by the broth microdilution method with 7H9 broth was performed in 96-well plates as described previously [36]. An initial load of 3 × 10^5^ bacteria was inoculated per well in Middlebrook 7H9 broth (Difco Laboratories, Detroit, MI, USA) at final volume of 200 µL. A range of 20–48 µg/mL and 10–80 µg/mL of IP-1 was tested for H37Rv and CIBIN99 strains, respectively. The cultures were incubated for 7 days at 35 °C and agitation at 70 rpm. Bacteria proliferation was determined by a colorimetric assay using Cell Titer 96^®^ Aqueous (Promega, Madison, WI, USA) and by measuring the absorbance at 492 nm (Epoch2, USA). To confirm the colorimetric results, bacteria were diluted in 7H9 medium and colonies were cultured and counted in Middlebrook 7H10 agar after 14 and 21 days.

To determine the bacillary loads in Mø treated with IP-1 peptide, the macrophage cell line J774A1 (ATCC TIB-67) were co-cultured for 4 h with *M. tuberculosis* H37Rv or clinical isolate CIBIN 99 (MDR) strains at a multiplicity of infection (MOI) of 5:1. Next, the cells were washed 3 times with fresh RPMI 1640 medium (Invitrogen Life Technologies, USA) plus antibiotic-antimycotic (Sigma P4333, USA) to remove unphagocytosed bacteria. The cells were subsequently treated with 8 to 32 µg of peptide IP-1 dissolved in 100 μL of vehicle and were incubated for 5 days. The cells were lysed with 0.1% SDS in 7H9 broth and after 10 min BSA 20% was added in broth 7H9 and the CFUs were determined by plating 10-fold serial dilutions onto Middlebrook 7H10 agar media supplemented with OADC. CFUs were counted after 2–3 weeks of incubation at 37 °C in 5% CO_2_. The results are expressed as the mean of three independent experiments. Statistical analysis was performed using one-way analysis of variance (ANOVA) for multiple observations.

### 2.13. Transmission Electron Microscopy

The subcellular alterations induced by IP-1 in infected and non-infected macrophages were evaluated using transmission electron microscopy. Briefly, infected and non-infected macrophages incubated with IP-1 at 8 µgr concentration were prepared for transmission electron microscopy (TEM) by pelleting the different cell preparations through centrifugation for 1 min/6000 rpm. The macrophages were fixed in 1% glutaraldehyde dissolved in 0.1 M cacodylate buffer (pH 7); postfixed in 2% osmium tetroxide; dehydrated with increasing concentrations of ethanol, and gradually infiltrated with Epon resin (Pelco). Thin sections were contrasted with uranyl acetate and lead citrate.

For morphometry, 30 cells from each condition were random selected at day one of incubation with IP-1 peptide and photographed at 40,000× magnification, and then the total number of phagosomes, autophagosomes, and lysosomes were counted in each experimental group.

For autophagosome confirmation, immunoelectronmicroscopy was used. Briefly, the macrophages were fixed in 4% paraformaldehyde in 0.2 M Sörensen’s buffer; the samples were dehydrated with different concentrations of ethylic alcohol and infiltrated with LR-White hydrosoluble resin (London Resin Co., Hampshire, UK). Sections that were 60 to 80 nm thick were placed on nickel grids. The grids were incubated overnight at 4 °C with specific polyclonal rabbit anti-LC3 (Novus Biologicals, Littleton, CO, USA) antibodies. After rinsing with PBS, the grids were incubated for 2 h at room temperature with goat anti-rabbit IgG (Sigma-Aldrich) conjugated to 5-nm gold particles (Sigma-Aldrich) and diluted 1:20 in PBS. The grids were contrasted with uranyl acetate (Electron Microscopy Sciences, Fort Washington, PA, USA) and examined with an Technai FEI electron microscope.

### 2.14. Experimental Model of Progressive Pulmonary Tuberculosis

The drug sensitive strain H37Rv (strain donated by the University of Stanford, Stanford, CA, USA) and MDR clinical isolate CIBIN99 was grown in Middlebrook 7H9 broth (Difco laboratories, USA) supplemented with Middlebrook OADC enrichment media (Difco laboratories, USA) and 0.05% Tyloxapol at 35 °C and 70 rpm. Mid-log phase culture was recovered and washed 3 times with sterile saline solution and stored at −80 °C until use. To determine the concentration of bacteria per ml, serial dilutions were made in 7H9 broth and they were cultured in Middlebrook 7H10 agar and counted 14 and 21 days later. The experimental model of progressive pulmonary TB has been described in detail elsewhere [37,38]. Briefly, male BALB/c mice aged 6–8 weeks were anaesthetized in a gas chamber using 0.1 mL per mouse of sevoflurane (Abbott Laboratories, Chicago, IL, USA), and each mouse was infected by endotracheal instillation with 2.5 × 10^5^ live bacilli in 100 μL saline solution, 0.02% tyloxapol. The mice were maintained in the vertical position until they underwent spontaneous recovery. Infected mice were maintained in groups of five in cages fitted with micro-isolators. All procedures were performed in a biological security cabinet at a Biosafety level III facility. The animal work was performed in accordance with Mexican national regulations on Animal Care and Experimentation (NOM 062-ZOO-1999) and according to the guidelines and approval of the Ethical Committee for Experimentation in Animals of the National Institute of Medical Sciences and Nutrition (INCMNSZ) in Mexico City, permit number CINVA 1825 PAT-1825-16/18-1, May 2016.

After 60 days of infection, the mice were arbitrarily divided into treated and control groups. The treated group was tested with a dose of 8 µg of the IP-1 peptide dissolved in 50 µL of saline solution and administered by intratracheal route every other day for one months, while the control group received only the vehicle solution. The efficiency of the treatment was determined by quantifying the lung bacillary loads by assessing colony forming units (CFUs) and the extent of tissue damage by histopathology and automated morphometry. To investigate the protective immune response induced by IP-1, lungs from H37Rv and MDR infected mice were used to determine the gene expression of TNFα and IFNγ by RT-PCR.

For the determination of bacillary loads and histopathology, immediately after the mice were euthanized by exsanguination under anesthesia with intraperitoneal pentobarbital, the lungs were removed; the right lung was immediately frozen by immersion in liquid nitrogen and used for CFU determination, while the left lung was perfused for histopathology analysis. For CFU determination, the frozen lungs were disrupted using 2 mm zirconia beads in tubes with 1 mL of PBS containing 0.05% tween in the Fast Prep24 MP Systems (MP Biomedicals, Irvine, CA, USA). Four dilutions of each homogenate were spread onto duplicate plates containing Bacto Middlebrook7H10 agar (Difco laboratories, USA) enriched with OADC. The incubation time of the plates was 14–21 days, and data points are the means of six animals. For the determination of tissue damage (pneumonia) by histomorphometry, the left lung was perfused via trachea with 100% ethanol and immersed for 24 h in the same fixative. Parasagital sections were taken through the hilus, and these were dehydrated, embedded in paraffin, and histological sections stained with hematoxylin and eosin were obtained. The percentage of lung area affected by pneumonia was measured using a Leica Q-win Image Analysis System (Cambridge, UK). The measurements were performed in a blind manner, and data are expressed as the mean of three animals ± standard deviation (SD). For the determination of the expression of protective cytokines by RT-PCR, left lung lobes from three different mice per group were used to isolate mRNA using the RNeasy mini kit (Qiagen, Hilden, Germany). The quality and quantity of RNA were evaluated through spectrophotometry (260/280) and on agarose gels. Reverse transcription of the mRNA was performed using 5 µg RNA, oligo (dT), and the Omniscript kit (Qiagen). Real-time PCR was carried out using the 7500 real-time PCR system (Applied Biosystems, Foster City, CA, USA) and Quantitect SYBR Green Master Mix kit (Qiagen). Standard curves of quantified and diluted PCR product, as well as negative controls, were included in each PCR run. Specific primers were designed using the program Primer 359 Express (Applied Biosystems) for the following targets: RPLP0 housekeeping gene F: 5′-CTCTCGCTTTCTGGAGGGTG-3′; R: 5′-357 ACGCGCTTGTACCCATTGAT-3′, IFNg F: 5′-GGTGACATGAAAATCCTGCAG-3′; R: 5′-CCTCAAACTTGGCAATACTCATGA-3′, TNFa F: 5′-TCGAGTGACAAGCCTGTAGCC-3′; R: 5′-TTGAGATCCATGCCGTTGG-3′. Cycling conditions used were as follows: initial denaturation at 95 °C for 15 min, followed by 40 cycles at 95 °C for 20 s, 60 °C for 20 s, and 72 °C for 35 s. Quantities of the specific mRNA in the sample were measured according to the corresponding gene-specific standard curve. The mRNA encoding RPLP0 was used as an internal invariant control to normalize the expression of TNFα and IFNγ genes. The results are shown as the relative expression in comparison to the control group.

## 3. Results

### 3.1. Cytotoxicity of IP-1 at High Dosages

To test the hypothesis that IP-1 has the dual function of inducing autophagy and AMP, we first determined the sub-lethal doses to be used in mammalian cells. We quantified cytotoxicity by trypan blue exclusion on mouse embryonic fibroblasts (MEFs, primary cells) and HEK293T (cell line) cells treated with IP-1 (see see Materials and Methods section). We observed a different response between the cells assayed. The less sensitive cells were HEK293T with 75% of surviving cells at 50 μM of IP-1 (see Appendix A). MEF cells were more sensitive to IP-1 showing 32% of surviving cells at 50 μM of IP-1 (see Appendix A).

DNA fragmentation is a feature of both apoptosis and necrotic cell death, and this degradation is detected by the terminal deoxynucleotidyl transferase dUTP nick-end labeling (TUNEL) assay [39,40]. We studied IP-1 toxicity in both MEF and HEK293T cells. According with trypan blue exclusion results, MEF cells were TUNEL positive starting at IP-1 concentrations 20 μM (see Figure 1A). On the contrary, HEK293T cells were TUNEL negative even at 50 μM (see Figure 1B). This observation suggests that MEFs are dying at doses of IP-1 sublethal for HEK293T cells.

The cytotoxicity of IP-1 might be the consequence of the amphipathic character of the peptide binding to and destabilizing the cell membrane, as we have previously confirmed in artificial membranes [27]. To test this, varying numbers of HEK293T cells were exposed to 50 μM of IP-1; only HEK293T cells were used in this assay because these do not develop contact inhibition, a feature that may affect the interpretation of our results, as we will discuss (see Discussion section). If the peptide binding to membrane were relevant for its toxicity, we would expect that larger number of cells would require larger amount of peptide to observe the cytotoxic effect. The mean effect of IP-1 on cell death remained constant independently of the initial number of cells exposed to IP-1 (see Appendix A). Therefore, IP-1 toxicity does not depend on the amount of available cell membrane.

Since DNA breaks also occur in mechanisms of cell death other than apoptosis, such as necrosis, we looked for the presence of phosphatidylserine on the outer leaflet of the plasma membranes, as it is a marker of apoptosis. Annexin V binds strongly to phophsatidylserine, and hence, it is a useful marker of apoptosis. HEK293T or MEF cells exposed to different concentrations of IP-1 or staurosporine and etoposide 2 μM as positive control of apoptosis. We observed that only MEF cells were Annexin V positive, although both cell types internalized propidium iodine indicating the cell membrane integrity was lost (see Figure 2). Together these results indicate IP-1 induced necrotic cell death on HEK293T cells, and apoptosis and necrosis on MEF cells at concentrations higher than the one used to induce autophagy (10 μM).

### 3.2. Respiratory Function of Cells Exposed to IP-1

To test for the effect of IP-1 on mitochondrial function, we evaluated the respiratory efficiency of cells exposed or not to IP-1. HEKT293T cells were selected for these experiments because the respiratory rate (here represented as slope of respiration) was detectable by an oxymeter (see Materials and Methods section). It is important to note that small changes in slopes in fact represent large changes in respiration rates; for instance, a slope of 1 changing to 2 represents doubling the respiration rate. First, we noted that cells in basal conditions presented slopes mainly from 0 to −2, indicating that these cells consumed small amounts of oxygen over time. We observed that HEK293T cells had a similar respiratory rate in the presence or absence of IP-1. In addition, for many individual experiments the respiration rates in the presence of IP-1 were identical (see Appendix A). For comparison, cells exposed to cyanide (Cn) displayed rates of respiration close to zero, confirming that respiratory was inhibited; in the case of cells exposed to a decoupling agent (FCCP) the respiration slope is mainly in −5, confirming the acceleration of respiration rate induced by this compound. ANOVA analysis showed that these experiments had significantly different mean values (*p* = 0.01), particularly treatments with FCCP and Cn. Post hoc Tukey HSD test confirms that the null hypothesis is rejected for most of these treatments at 95% confidence interval (see Materials and Methods section).

To confirm the null effect of IP-1 on mitochondrial function in mammalian cells, we used JC-1, a dye that stains mitochondria as a function of its membrane potential [41]. Briefly, at low mitochondrial membrane potential (a condition prevalent in mitochondria with high rates of respiration), JC-1 does not aggregate (the intensity of fluorescence is low and is measured at 525 nm), while at higher membrane potential (a condition prevalent in mitochondria with low rates of respiration) JC-1 aggregates (the intensity of fluorescence is high and is detected at 590 nm). To measure mitochondrial function in individual cells, we used a flow-cytometer coupled to a multi-photonic microscope and counted the number of events/cells with high and low fluorescence intensity to estimate the respiration of individual cells (see Materials and Methods section). We observed that 18% of cells grown in normal conditions presented low mitochondrial membrane potential, indicating that few cells have active mitochondria, which correlates with measured respiration rates. On the other hand, 97% of cells treated with the uncoupling agent FCCP showed low membrane potential, correlating with the high rate of oxygen consumption measured in these cells. We observed no differences in JC-1 fluorescence in cells exposed to 0 or 10 μM IP-1 after 15 or 60 min of exposure. In contrast, IP-1 at 50 μM, the high and low membrane potential events tend towards 50% (see Appendix A), indicating that IP-1 at 50 μM induced a partial membrane potential alteration. Based on these experiments, we concluded that IP-1 affects mitochondrial membrane potential at 50 μM yet without significantly affecting the respiration rate.

### 3.3. No Lipid Changes in Response to IP-1

Autophagy, while catabolic in essence, requires energy at different steps [42]. Since we did not detect any mitochondria disfunction, we studied if the amphipathic character of IP-1 can destabilize membranes from other organelles and hence induce autophagy to degrade lipids at the damaged membranes. Lipidome changes induced by AMPs on immature dendritic cells have been detected by optical microscopy and IR microspectroscopy [43]. HEK293T cells were exposed to rapamycin or substance P (SP) for 6 h as positive controls of autophagy; statistical analysis performed on IR spectrum of live cells (Appendix A) (hierarchical cluster analysis—Appendix A, and principal component analysis Appendix A) treated with IP-1, rapamicyn, or substance P in the infrared region characteristic to the lipids due to the CH_2_ and CH_3_ stretching vibrations showed no significant differences. There were no discernible clusters in data due to the treatment even when the variance explained by the two first components is 75.9%, hence, we concluded that the differences in treatment do not constitute a significant change in macromolecule composition within cells. Finally, we quantified by mass spectroscopy the non-esterified fatty acids extracted from HEK293T cells exposed to 10 or 50 μM of IP-1 for 6 h and could not detect any significant difference compared with cells not exposed to IP-1 (see Appendix A).

### 3.4. ATP Quenching by IP-1

Considering the cationic character of IP-1, we hypothesized that this peptide could bind ATP, and consequently induce both autophagy and cell death in a dose-dependent manner, i.e., small amounts of IP-1 may induce autophagy while larger amounts of IP-1 may induce cell death. To test this hypothesis, we conducted a functional assay where the luciferase reaction was used to monitor the ATP quenching ability of IP-1. The luciferase reaction depends on the luciferin cofactor and produces one photon per each ATP molecule and can be followed by photometry [44]. We incubated ATP with and without IP-1 and then added this mixture to the luciferase enzyme to quantify the amount of ATP available in the media; the reaction of luciferase with ATP is instantaneous (less than 3 s) so, this essay may reveal the ability of IP-1 to quench ATP. We observed that IP-1 inhibits the luciferase reaction (*p* = 5.61 × 10^−30^ with a Wilcoxon test) at equimolar concentration with ATP (see Appendix A). To further validate the binding of IP-1 to ATP, we performed an isothermal titration calorimetric assay (see Materials and Methods section section) and determined the affinity constant of IP-1 for ATP to be 440 μM (see Appendix A). Finally, we determined intra and extra cellular amounts of ATP in HEK293T cells exposed to IP-1 (10 and 50 μM). HEK293T presented a measurable response to IP-1 in terms of the ATP levels: (i) at 10 μM, intra and extracellular ATP concentration was not altered; (ii) at 50 μM, extra cellular ATP behaved similarly than at 10 μM, but intra cellular ATP concentration decreased during the experiment (see Figure 3). Since no significant effect on respiration was observed at 50 μM of IP-1, we conclude that the reduction in intracellular ATP levels is the consequence of IP-1 binding ATP.

### 3.5. IP-1 Induces Autophagy and Has Cytotoxic Effects on Macrophages

The ability of the IP-1 to induce autophagy was assessed in two mammalian cell types: HEK293T and macrophages. We detected the abundance of autophagosomes in HEK293T cells by two methods. First, we used Cyto-ID^®^ staining, a fluorescent cationic amphiphilic compound that accumulates in autophagosome vesicles. IP-1 increased the number of HEK293T cells with more than 6 detectable autophagosomes per cell compared with cells not treated with IP-1 (see Appendix A). Autophagy induction was dose dependent (see Appendix A). Second, we followed LC3-II lipidation, which only occurs during autophagosome formation (see Appendix A). We demonstrated that IP-1 induced the autophagic flux by detection of green or red fluorescent proteins fused to LC3 (see Appendix A), as well as by estimating the degradation of p62/SQST1 by Western blot (see Appendix A). Finally, we compared the abundance of autophagosomes with or without Chloroquine, an inhibitor of autolysosomes formation (see Appendix A). Together, these results show that IP-1 induces autophagy in HEK293T cells in a dose-dependent manner.

To test for the biological relevance of this activity in the context of MTB infections, we exposed macrophages to IP-1. Macrophages are infected by MTB during the establishment of pulmonary tuberculosis and these cells respond by activating autophagy, but MTB inhibits this process; further activation of autophagy reduces MTB survival within infected macrophages [45]. Hence, we first detected the concentration of IP-1 required to observe 50% of cell death in a cell line of macrophages (J774.1); we observed that below 16 μg/100 μL (64 μM) of IP-1 killed about 25% of the macrophages (see Figure 4 top panel, *p* < 0.001; a representative image can be seen in Appendix A). Consequently, autophagosome formation by J774.1 cells exposed to 8 μg/100 μL (32 μM) of IP-1 was measured using CytoID as reported above and observed as much autophagosomes as those observed in macrophages exposed to rapamycin (positive control of autophagy) and two times more than macrophages not exposed to any autophagy induction (see Figure 4 bottom panel; *p* < 0.001).

To investigate whether IP-1 peptide could have access to MBT inside the autophagosomes, and hence to be able to execute its AMP activity concomitant with the induction of autophagy, we analyzed the intracellular distribution of a fluorescently labeled IP-1 peptide and simultaneously detected autophagosomes with CytoID staining. As shown in Appendix A, IP-1 was found inside of autophagosomes in macrophages non infected, as well as in cells infected with MTB strain H37Rv sensitive to the first line of antibiotics (H37Rv) and a clinical isolate (CIBIN99) that is resistant to those antibiotics (MDR). To further evaluate the ability of IP-1 to induce autophagy in macrophages, the ultrastructural and morphometry studies showed that IP-1 incubation after infection of macrophages with mycobacteria induced more autophagosomes (Figure 5A–C), which was confirmed by immune-labeling to LC-3 using colloidal gold, and significantly decreased the bacillary loads after one, three and five days of infection, when compared with infected cells without IP-1 peptide (Appendix A). Direct observation in chamber slides of these infected macrophages incubated with 3.21 μM (8 μg/mL) of IP-1 during 5 days showed 55.8% cell survival with many activated cells (large cells with big nucleus), while incubation with 6.42 μM (16 μg/mL) decreased the cell survival to only 5% (data not shown). This observation is consistent with previous observations indicating that in TB autophagy confers protection while necrosis disseminates the disease.

During macrophage activation there is autophagy activation and cytokine production [46], tumor necrosis factor alpha (TNFα) being one of the most significant protective cytokines. Thus, 5 × 10^6^ infected macrophages plated in 24-well dishes were incubated during 24 h with or without IP-1 and supernatants were collected to assess TNFα production determined by ELISA according to published method [47]. There was a dose response effect, being the IP-1 concentration of 8 µg/100µL the best inducer of TNFα production, while the highest IP-1 concentration (32 μg/100 μL) induced lower TNFα production, as a consequence of the increased number of dead cells at that concentration (Figure 5E). Thus, low or mild concentrations of IP-1 induces autophagy and the production of TNFα in macrophages, which is one of the most protective cytokines against MTB infection.

### 3.6. Antibiotic Activity of IP-1 against M. Tuberculosis

We used different concentrations of IP-1 to test for its direct antibacterial activity against MTB in vitro; using two strains of MTB, one sensitive to the first line of antibiotics (H37Rv) and a clinical isolate (CIBIN99) that is resistant to those antibiotics (MDR). Our results showed that IP-1 reduced the viability of H37Rv by 98% at 200 μg/mL, with an IC-50 of 99.27 μM (247.25 μg/mL), while for the MDR strain was less resistant (IC-50 92.66 μM or 230.78 μg/mL), yet susceptible to be killed by IP-1 at higher concentrations (see Figure 6, left panels). The MIC positive control was amikacin (AMK), an antibiotic for which this MDR strain is highly susceptible, and for drug-sensible H37Rv strain was isoniazid (INH) that is the most efficient primary antibiotic. As a negative control, we used a synthetic peptide derived from arachnid AMP (P2 in Figure 6) that has slight antimicrobial activity against Gram- bacteria and without any autophagy induction effect [48].

The bactericidal activity of the IP-1 peptide was tested against progressive pulmonary tuberculosis induced by drug susceptible H37Rv strain or MDR clinical isolate using an experimental model in BALB/c mice. By intratracheal route 8 µg of IP-1 was administered every other day for one month, starting after two months of infection, when in this model active progressive disease is in advanced stage. Treating infected animals with H37Rv strain caused a significant reduction of the lung bacillary load (greater than 80%) in comparison with the negative control (infected mice simply treated with saline solution) (Figure 7 top right panel). Treated mice showed a reduction in the bacillary counts (see Figure 7A) and slight reduction of pneumonia (see Figure 7B). Infected mice with MDR strain treated with the same dose and administration route of IP-1 also showed significant decrease of CFUs (Figure 7 bottom right panel). This result correlated with the histological changes; treated mice showed a decrease in tissue damage (pneumonia) compared to the control mice (Figure 7C,D) and with the expression of protective cytokines, such as IFNγ and TNFα that was higher in drug sensitive or MDR infected mice treated with IP-1 (see Appendix A). The images in Figure 7 are one representative experiment.

## 4. Discussion

The discovery of novel antibiotics represents an opportunity to treat emergent and resistant infectious agents and at the same time constitutes a tool to discover/study cellular mechanisms. In the present work, we provide evidence that IP-1, a peptide designed to kill bacteria, activates autophagy in mammalian cells (macrophages and HEK293T cells). Such a combination of activities would constitute a new class of antibiotic that may be effective in treating infectious diseases. We provide evidence that IP-1 kills MTB at a dose (8 μg/100 μL or 32 μM) that is not toxic to macrophages and induces autophagy. Hence, our data suggest that IP-1 may be useful in the treatment of MTB infections; for instance, by inducing autophagy in infected macrophages may reduce the risk of autophagosome’s rupture hence preventing the release MTB spreading to new cells [49] and by directly killing MTB.

In this work, we further explore the mechanism of action of IP-1. In a previous work, we tested IP-1 in microbial eukaryotic cells (*Saccharomyces cerevisiae*) and noted that IP-1 altered their respiration and induced the swelling of isolated yeast mitochondria [27]. Here, we describe that IP-1 does not alter mitochondrial potential in mammalian cells nor kills mammalian cells at low concentration (10 μM), but at higher concentration (50 μM) IP-1 is toxic for both yeast and mammalian cells [50]. Thus, IP-1 toxicity does not involve mitochondrial function. In agreement with this idea, we have shown that IP-1 kills yeast cells lacking mitochondria DNA [27] and more recently reported that IP-1 kills yeast cells grown in either respiratory (mitochondria competent) or fermenting (mitochondria incompetent) media [51].

For IP-1 to reach mitochondria, this peptide must be internalized. In that sense, it has been observed that AMPs that are internalized by mammalian cells may damage mitochondria membrane because of their similarity with bacterial membranes [52]. We have shown that IP-1 penetrates both yeast and mammalian cells by a mechanism independent of endocytosis [27], yet our current results indicate that such internalization does not modify mitochondrial function in mammalian cells. Since the killing induced by IP-1 in both yeast and mammalian cells does not involve mitochondrial function, it is unlikely that IP-1 may target mitochondria in yeast cells either. Further work is required to establish the cellular sorting of IP-1 upon internalization.

Our results also show that at 50 μM of IP-1, the ATP levels were more affected in the intracellular space than in the extracellular medium. This result is in agreement with the relatively large affinity-binding constant of IP-1 for ATP, since most ATP is inside cells. However, at 10 μM IP-1 induces autophagy, yet no ATP reduction is observed in cells, suggesting that ATP may be reduced at some particular cellular location, but not globally; further experiments are required to validate this idea. Binding ATP could affect the acidification of autophalysosomes and consequently inhibit autophagy flux; however, our results show that IP-1 induces the full autophagy flux. Thus, the ATP binding by IP-1 may take place somewhere else than the autophalysosome. Recent reports claim that extracellular ATP is controlled by the intracellular production of ATP [53]; yet, at 50 μM IP-1 reduces the ATP level inside cells without altering the ATP levels outside. This may be explained by our observation that IP-1 kills approximately 30% of cells at 50 μM, hence, these cells may not consume extracellular ATP and consequently maintain the extracellular ATP levels. Further in agreement with these observations are our results that IP-1 did not alter mitochondrial potential nor the respiration rate, and consequently, did not promote any change in ATP production. Extracellular ATP has been shown to play a role as a signaling molecule in inflammation, autophagy, tissue damage, and mortality [54], and it also plays a signaling role in yeast cells [55]. Increasing the secretion of ATP regulates P2RX7 receptor to induce autophagy and cell death [20,56]. Hence, the mechanism of action of IP-1 cannot be explained by this single receptor based on our current results. In any case, IP-1 may present multiple outcomes by targeting a single ubiquitous target; in the present work we showed that ATP is a target that may explain cell death and autophagy induction, yet it remains to be elucidated whether this single target explains the increased membrane permeability in HEK293T cells or the TNFα increase production in macrophages for instance. Our results on multiple outcomes for binding ATP by IP-1 in treating MTB infection provide an alternative to interpret the role of other AMPs on treating infections. For instance, LL-37 was shown to induce autophagy dependent on the P2RX7 receptor by inhibiting the action of this receptor [19], which as noted above is known to activate autophagy by sensing ATP. It would not be surprising that LL-37 would bind ATP since diverse AMPs interact with ATP [57].

A recent review on all AMPs reported to kill MTB (MDR or not) describes 3 peptides that act through ATP: Lassomycin cyclo targets ATP-dependent protease ClpC1P1P2, and Trichoderin A or B inhibits ATP synthase [58]. Only Lassomycin cyclo has been shown to kill MDR MTB and display no toxicity against mammalian cells at the concentration reported to kill MTB; this peptide is cyclic, which makes it stable in biological conditions, but the cost of synthesis is higher. On the other hand, Tricoderin A or B are both aminolipopeptides. All these three peptides were derived from natural sources. Hence, all these peptides are not linear and require special conditions for their synthesis. IP-1 synthesis is compatible with standard chemical and biotechnological procedures. In the same review [58], 16 AMPs with activity against MDR MTB are described; none of the mechanisms of action of these 16 AMPs combine autophagy with an antibacterial activity. Thus, IP-1 provides a novel polypharmacologic molecule to treat TB that is amenable to standard procedures for large-scale peptide synthesis.

With more than 1.3 million deaths annually in the world, TB is currently the leading cause of death by a single infectious agent [59]. Although TB can be cured by chemotherapy, this treatment usually requires four specific drugs and 6 months of therapy in humans, which produces significant compliance problems, disease recrudescence, and the rise of MDR strains, whose treatment requires more than eight antibiotics administered during two years producing higher toxicity and cost. In mice and humans, mycobacterial infections are controlled by the activation of macrophages through type 1 cytokine production by T cells. IFNγ and TNFα are essential for this process because they promote macrophage activation and induce the production of reactive oxygen and nitrogen species. The intracellular receptors NOD1 and NOD2 recognize mycobacterial compound, such as MDP that triggers autophagy and induce the production of pro-inflammatory mediators, such as TNFα in alveolar macrophages [46,60]. Our results showed that IP-1 induced high production of TNFα by MTB infected macrophages, the same response plus high production of IFNγ was determined in the lungs of BALB/c mice infected with drug sensible or resistant MTB treated with IP-1. The high production of these cytokines could participate in not only activating macrophages, but also promoting autophagy [61], and increased autophagy in coexistence with NOD receptors activation can produce a higher secretion of protective cytokines and antimicrobial peptides such as LL-37 [46,60]. Thus, the efficient in vivo anti-TB effect showed by IP-1 derived from multiple activities suggest that it can be used as adjuvant to improve chemotherapy against drug susceptible and resistant TB.

In summary, IP-1 is shown to be effective in treating both sensitive and MDR MTB infected mice. This therapeutic effect of IP-1 is associated with its ability to kill MTB at a dose where autophagy is induced in mammalian cells, including macrophages; at higher doses, IP-1 induces cell death. Both activities may be explained by the ability of IP-1 to sequester ATP.

## 5. Patents

The use of peptide Iztli 1 as an antimicrobial agent is protected in Mexico under patent number MX 350389 B.

## Figures and Tables

**Figure 1 pharmaceutics-12-01071-f001:**
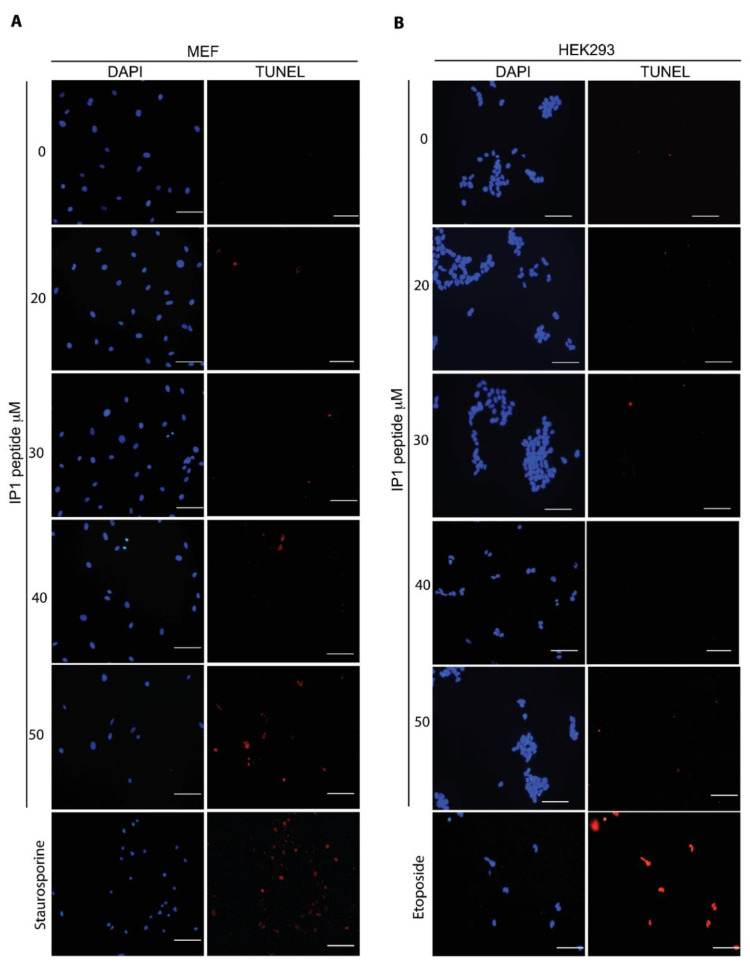
IP-1 induces DNA fragmentation in mammalian cells. Mouse Embryonic Fibroblasts (MEFs) (**A**) or HEK293T cells (**B**) were incubated for 6 h with the indicated concentrations of IP-1 peptide. As positive controls, MEFS were treated with 2 μM Staurosporine (**A**) and HEK293T cells were treated with 400 μM etoposide for 2 h. Notice an increasing number of TUNEL-positive cells, which indicates DNA fragmentation, in cells treated with increasing concentrations of IP-1; the experiments were conducted once and were validated with annexin V staining (see below). Nuclei were labeled with DAPI. Scale bar represents 100 μm.

**Figure 2 pharmaceutics-12-01071-f002:**
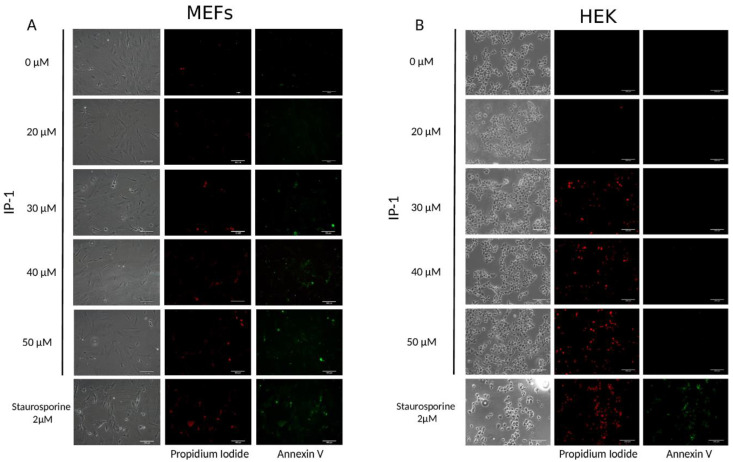
IP-1 induces apoptosis at low doses in MEF cells and necrosis at higher doses in HEK293T cells. MEF (**A**) and HEK293T (**B**) cells were treated with IP-1 at the indicated concentrations or with 2 μM Staurosporine for 2 h as positive control. Then, the cells were stained with Annexin V-FITC in order to detect PS exposed to the outer layer of the cell membrane, an event related to apoptosis induction. Additionally, cells that have lost membrane integrity also incorporate propidium iodide, showing red fluorescent staining. As shown in (**A**), starting at 30 μM of IP-1, MEF cells present Annexin V or propidium iodide staining, unlike to HEK293T cells that only incorporated propidium iodide. Experiments were conducted twice; the images are representative. Scale bar represents 100 μM.

**Figure 3 pharmaceutics-12-01071-f003:**
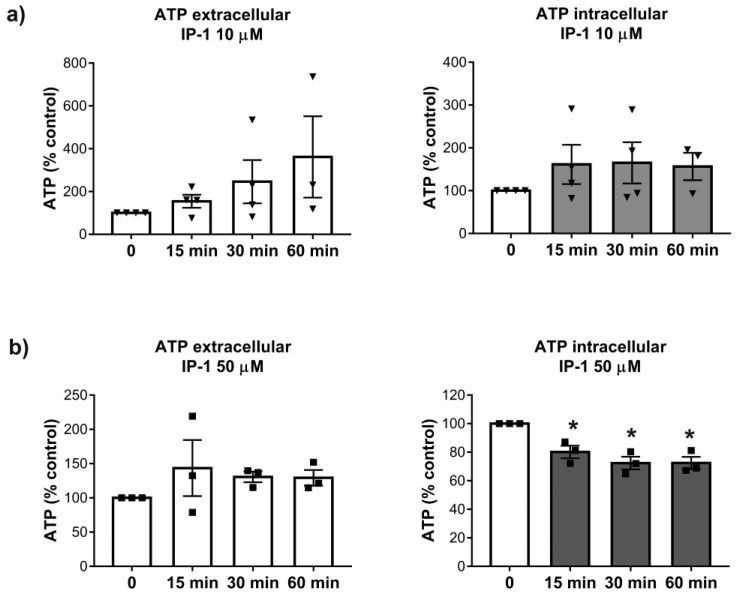
Intra and extracelullar ATP levels of HEK293T cells treated with PI-1. Upper panel (**a**) shows the extra (left) and intracellular (right) ATP levels of HEK293T cells exposed at IP-1 10 μM. Lower panel (**b**) shows the extracellular (left) and intracellular (right) ATP levels of HEK293T cells exposed at IP-1 50 μM. The image was generated using R package. Each plot represents the normalized ATP levels of 4 different experiments on HEK293T cells (see Materials and Methods). Each symbol (Triangles, squares) represent an independent determination. Asterisk represents statistical significance *p* < 0.05 with respect to measurements at time 0.

**Figure 4 pharmaceutics-12-01071-f004:**
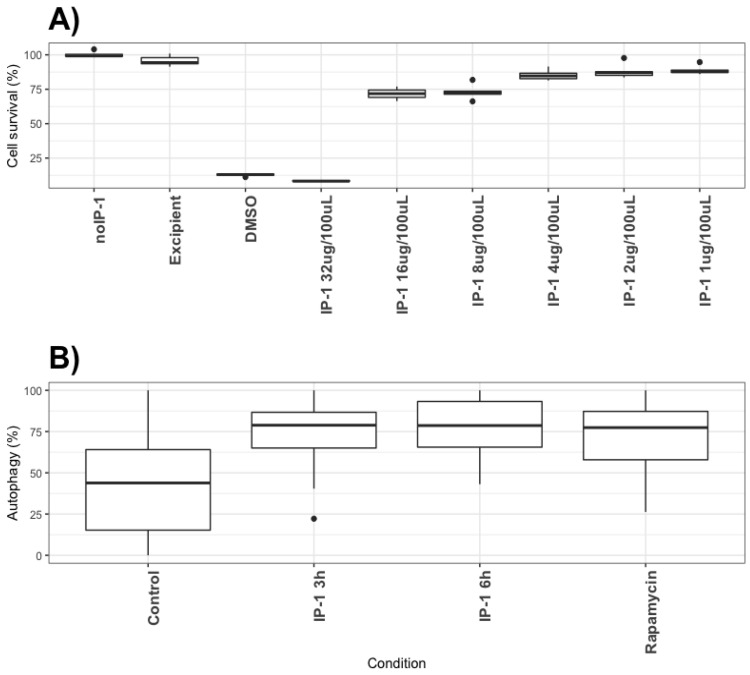
Cytotoxicity and autophagy induction of IP-1 on macrophages. Top panel: (**A**) Percentage of macrophages (J774.1) staining with violet dye, as an indicator of cell survival (see Materials and Methods section). noIP-1 and Excipient shows percentage of staining on cells without IP-1; DMSO is a toxic agent to macrophages and serves as positive control for dead cells; different concentrations of IP-1 were used (1, 2, 4, 8, 16, and 32 μM) to test for the survival of macrophages. The data were derived from at least 3 to 6 independent experiments. Bottom panel: (**B**) The percentage of cells showing green dots as indicative of autophagy as detected by Cyto-ID (see Materials and Methods section). Control are cells not exposed to IP-1, Rapamycin show the results of cells exposed to the autophagy-inducer rapamycin, and the autophagy detected at 3 and 6 h in the presence of IP-1 are shown. These data were derived from at least 30 photographs obtained from 3 independent experiments. The statistical comparison between groups was performed using ANOVA test and the *p*-values were obtained from a Post-hoc analysis with Tukey test (see [34]). The black circles represent outliers.

**Figure 5 pharmaceutics-12-01071-f005:**
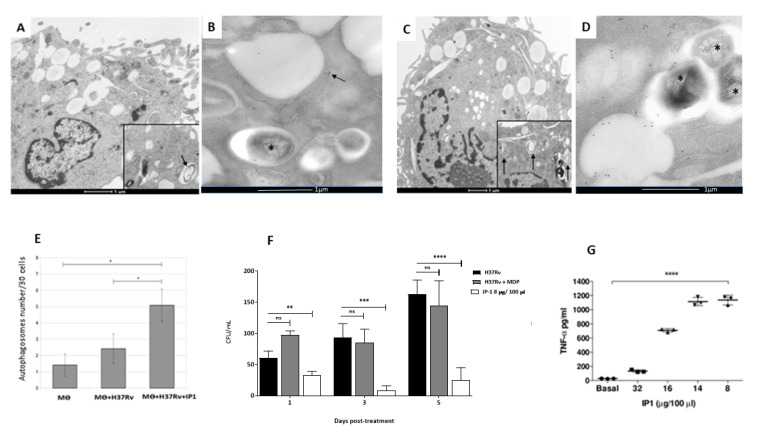
Effect of IP-1 on macrophages infected in-vitro with *M. tuberculosis* strain H37Rv. Representative electron microscopy micrographs of: (**A**) macrophage after one hour infection showing a phagocytosed bacteria and in the inset occasional double membrane vacuoles that correspond to autophagosomes (arrow). (**B**) Subcellular detection of autophagosomes by immunolabeling with specific antibodies against LC3, macrophage after 24 h of infection showing a phagosome with bacteria (asterisk) near to vacuoles/lysosomes surrounded by LC-3 detected by gold labeled antibodies (black dots, arrow). (**C**) Infected macrophage and incubated with IP-1 showing bacterial debris and in the inset several autophagosomes (arrows). (**D**) Infected macrophage incubated with LC-3 showing bacteria into phagosomes (asterisks) near to several LC-3 immunogold labeled vacuoles/lysosomes (black dots). (**E**) The morphometric study confirms significantly more autophagosomes (*y*-axis) in infected macrophages incubated with IP-1. (**F**) Determination of intracellular bacillary loads by colony forming units counts (CFU) in infected macrophages with mycobacteria strain H37Rv, after one, three and five days of incubation with MDP (muramil dipeptide control) or with 16 µg/100 µL of IP-1; all the studied times showed significant decrease of live bacillary counts. (**G**) Macrophages were infected with mycobacteria strain H37Rv, and after 24 h of incubation with the indicated concentration of IP-1 the supernatants were collected and used to determine TNFα by ELISA, low IP-1 concentrations (8 and 16 μg/100 μL) induced significant amount of TNFα, while larger IP-1 concentration (32 μg/100 μL) reduced its production. Each symbol represents one independent experiment. Each symbol (Triangles, squares, circles) represents one independent experiment. Asterisk represents statistical significance (* *p* < 0.05, ** *p* < 0.003, *** *p* < 0.002, **** *p* < 0.0005). ns: not significant.

**Figure 6 pharmaceutics-12-01071-f006:**
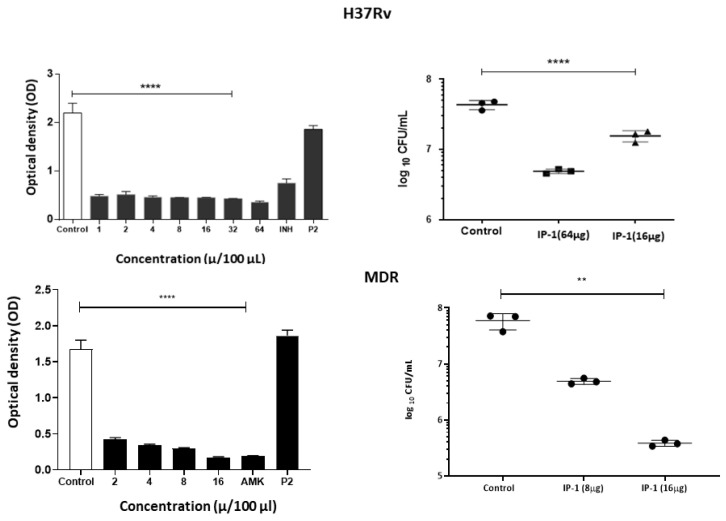
Effect of IP-1 against *M. tuberculosis* (top panel H37Rv, bottom panel MDR strain) in vitro. Left figures show the MICs determined by broth microdilution evaluated by a colorimetric assay using Cell Titer 96^®^ Aqueous, and right figures show viability of the bacteria by counting the colony-forming units after treatment with the indicated IP-1 concentrations; each symbol (Triangles, squares, circles) represent an independent determination. All the IP-1 concentrations showed significant activity against mycobacteria, being MDR strain more susceptible to IP-1 and similar to the MIC control that corresponded to amikacin (AMK), an antibiotic for which this strain is highly susceptible, and for drug-sensible H37Rv strain isoniazid (INH) that is the most efficient primary antibiotic. As negative control we include P2, a synthetic peptide derived from arachnid AMP at 64 μg. Each result is reported as the mean ± SD, asterisk represents statistical significance (** *p* < 0.003, **** *p* < 0.0005).

**Figure 7 pharmaceutics-12-01071-f007:**
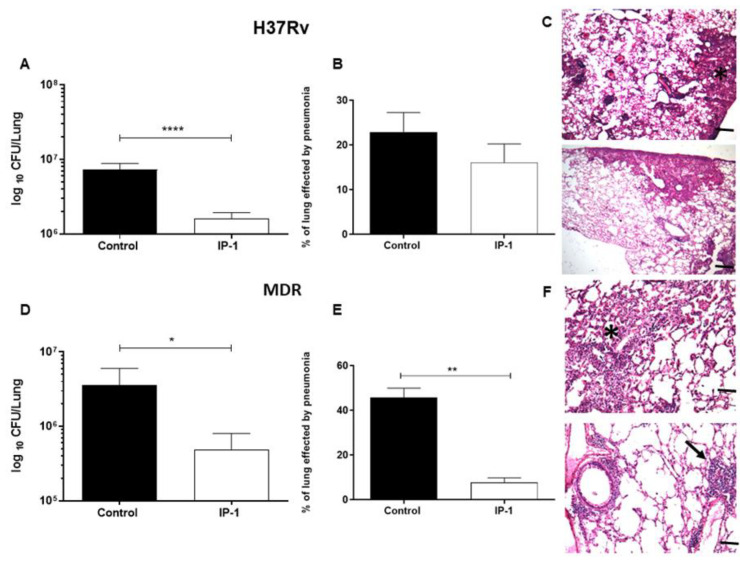
Effect of IP-1 treatment in BALB/c mice at 60 days post-infection with the drug-sensitive or drug resistant MTB strains. (**A**) Groups of mice infected with drug sensitive H37Rv strain were treated with 8 µg of IP-1 by intratracheal route each other day during one month, right lung was used to determinate bacillary loads by colony forming units (CFU), treated mice showed a significant decrease of live bacilli in comparison with control mice. (**B**) Left lungs were perfused with formaldehyde and used to determine pneumonia by automated morphometry, in comparison with control mice, a non-significant decrease of lung consolidation was seen in treated animals. (**C**) Representative micrographs of control non-treated mouse infected with H37Rv strain (top figure) shows similar lung surface area affected by pneumonia (asterisk) that in mouse treated with IP-1 (bottom figure). (**D**) The same treatment in mice infected with MDR strain showed a significant decrease of bacillary loads and lung surface affected by pneumonia. (**E**) Asterisk represents statistical significance (*p* < 0.05). (**F**) Representative micrograph of control non-treated mouse infected with MDR strain showing areas of pneumonia (asterisk, top figure), in comparison with MDR infected mouse treated with IP-1 that exhibits small patches of pneumonia (arrow, bottom figure) and perivascular inflammation (H/E staining, magnification 100×). The images correspond with one representative experiment. The bars in images (**C**) and (**F**) represent 200 μm. Asterisk represents statistical significance (* *p* < 0.05, ** *p* < 0.003, **** *p* < 0.0005).

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
