# Peer review of "Antimicrobial Peptide against Mycobacterium Tuberculosis That Activates Autophagy Is an Effective Treatment for Tuberculosis"

_pharmaceutics, 2020, doi:10.3390/pharmaceutics12111071_

Round 1
Reviewer 1 Report
Manuscript Number: pharmaceutics-940839
Antimicrobial peptide against Mycobacterium tuberculosis that activates autophagy is an effective treatment for tuberculosis
Erika et al. reports that an antimicrobial peptide, IP-1(KFLNRFWHWLQLKPGQPMY) designed to kill bacteria activates autophagy in HEK293T cells and macrophages. The synthetic peptide is shown to be effective in treating both sensitive and MDR MTB (mycobacterium tuberculosis) infected mice. This therapeutic effect of IP-1 is related to its ability to kill MTB at a dose that autophagy is induced in mammalian cells. The study also found that the mechanism of peptide is not mediated by altering mitochondrial integrity, but one of its products. IP-1 binds to ATP and it also decreased intracellular ATP concentration, and this ability of IP-1 to bind ATP may produce antimicrobial activity and autophagy that may eliminate MTB. In summary, IP-1 is the antimicrobial peptide that eliminates MDR MTB infection by combining four activities: reducing ATP levels, bactericidal activity, TNF-α secretion and autophagy activation. The findings are interesting and well-designed, organized, and descripted. But, several questions were occurred. I have minor questions.
- Please, result and discussion part should be described in detail, to increase reader’s understanding.
- There are minor typing errors in this paper, so please check the paper.
- High dose of IP-1 showed toxicity in many cells, and I wonder how about the toxicity in mouse?
- This article described the induction of autophagy by IP-1 increased the level of TNF-α or IFN-γ. It is better to add the description how to increase the level of TNF-α or IFN-γ by autophagy in discussion.
- Figure 4 showed IP-1 have a toxicity over 30uM in MEF and HEK293T cell, but 50uM of IP-1 was used in a measurement of ATP level (Figure 5). I wonder decrease of ATP intracellular level in Figure 5B is caused by the toxicity of IP-1.
- As it is not sufficient for explaining the cause of autophagy, Are 3.3 and 3.4 paragraph necessary?
- Are there any results by IP (intraperitoneal), IV (intravenous), or oral injection except the result by intratracheal route? Is it specific to lung? If the IP-1 applies to other organs except lung, what do you think?
- What do you think about half-life of IP-1 peptide?
[3.1. IP-1 induced autophagy in mammalian cells]
- The result 3.1 describes that IP-1 increased the number of HEK293T cells with more than 6 detectable autophagosomes per cell compared with cells not treated with IP-1 or an autophagy inducer (Trehalose), but Figure 1 seems that cells treated with Trehalose form autophagosomes than other cells treated with IP-1. Please describe in detail if it is not the error of sentence.
- To explain the Figure 2B, the 3.1 paragraph says that an alternative method to monitor autophagic flux is by following the degradation of the autophagy receptor p62/SQSTM1, but the paragraph of introduction regarding p62/SQSTM1 states that the abundance of p62/SQSTM1 is indicative of a functional autophagic flux. I am confused about this conflict, so please add the details in the paragraphs.
[3.2 Cytotoxicity of IP-1 at high dosages]
- By explanation of Figure 4, I understood IP-1 at high dosages causes necrosis and apoptosis, but Figure 4 does not describe the measurement of IP-1 at 10uM although Figure 4 paragraph seems to focus on the result of IP-1 at 10uM and 50uM. Why?
[3.5 ATP quenching by IP-1]
- To test hypothesis that small amounts of IP-1(10uM) may induce autophagy, the experiment was conducted by monitoring the ATP quenching ability of IP-1. But in Figure 5, Intra and extracellular ATP concentration of cells exposed to IP-1 at 10uM was not altered. How did autophagy happen at 10uM in spite of the constant ATP level?

Author Response
1. Please, result and discussion part should be described in detail, to increase reader’s understanding.
We appreciate the comments that helped us improved the description of our work. By attending the reviewers comments, we believe we have clarified the results and discussion sections of our work.
2. There are minor typing errors in this paper, so please check the paper.
We have double-check the document for typing errors and fixed them.
3. High dose of IP-1 showed toxicity in many cells, and I wonder how about the toxicity in mouse?
We did not run specific IP-1 toxicity assays in vivo. However, the lung histology of treated mice did not show any more tissue damage than pneumonia which was similar or lower in IP-1 treated animals than in the control non-treated mice group. In particular, we did not observed necrosis in the lungs of treated mice. Thus, it seems that IP-1 is not producing pulmonary damage at least at the IP-1 low dose that we used.
4. This article described the induction of autophagy by IP-1 increased the level of TNF-α or IFN-γ. It is better to add the description how to increase the level of TNF-α or IFN-γ by autophagy in discussion.
Mycobacterial compounds such as MDP are recognized and activate the intracellular receptors NOD1 and NOD2 inducing autophagy and secretion of pro-inflammatory cytokines and antimicrobial peptides (LL-37) by alveolar macrophages. The increase of autophagy mediated by IP-1 that promote higher bacterial killing could liberate more bacterial compounds that increase NOD activation, autophagy and cytokines and antimicrobial peptides secretion. This new information supported by two references are now added in the discussion.
5. Figure 4 showed IP-1 have a toxicity over 30uM in MEF and HEK293T cell, but 50uM of IP-1 was used in a measurement of ATP level (Figure 5). I wonder decrease of ATP intracellular level in Figure 5B is caused by the toxicity of IP-1.
Figure 4 (new Figure 2) shows that IP-1 at 30 microM permeabilizes HEK cells and figure S1 shows that 50 microM of IP-1 clearly reduces cell viability. That is the motivation to use that concentration in the ATP experiment. The ATP measurements were conducted 15, 30 and 60 minutes after IP-1 was added to the media, while cell survival was measured after 6 hours. In a previous work where we studied the internalization of IP-1 into different mammalian cells (https://www.ncbi.nlm.nih.gov/pmc/articles/PMC4031501/), we did not observe any toxicity during the first 60 minutes of exposure to IP-1. Thus, the decreased ATP concentration in Figure 5B (new Figure 3B) is not the consequence of cell dead.
6. As it is not sufficient for explaining the cause of autophagy, Are 3.3 and 3.4 paragraph necessary?
Paragraph 3.3 (IP-1 effect on respiratory function of cells) and 3.4 (lipid composition of cells exposed to IP-1) are presented to discard possible alternative mechanisms of action. We cited in the Introduction that autophagy may be induced by damaging mitochondria function, yet here we show that is not the case. We also described in paragraph 3.4 that amphipatic peptides such as IP-1 may alter membrane stability (for instance, we showed that MEF and HEK cellular membrane permeability were affected) and such perturbation could trigger autophagy, yet we showed that is not the case. These results led us to evaluate the ability of IP-1 to quench ATP and we showed it does both in vitro and in vivo. For these reasons, we thought it would be relevant for the readers to understand what led us to propose ATP quenching as the mechanism that triggers autophagy.
7. Are there any results by IP (intraperitoneal), IV (intravenous), or oral injection except the result by intratracheal route? Is it specific to lung? If the IP-1 applies to other organs except lung, what do you think?
We have used this mouse model of pulmonary TB for testing diverse antimicrobial peptides, and none of them showed antibacterial activity when were administrated by subcutaneous or intraperitoneal route, so we took for granted that IP-1 should be administrated by intratracheal route and no other routes of administration were tested.
8. What do you think about half-life of IP-1 peptide?
This is an interesting question. We have measured that this peptide is stable for up to 2 weeks in water at temperatures ranging around 15-30 Celsius degrees (unpublished results). However, if we add this peptide to a cell culture, which is at 37 Celsius degrees, we cannot detect any activity after 15 or 30 minutes (unpublished results). Since we have already reported that the peptide is attached to cell membranes in the first 15 minutes of exposure (https://www.ncbi.nlm.nih.gov/pmc/articles/PMC4031501/), we believe that either after 15 or 30 minutes, the active peptide is already attached to cells and the rest could have been inactivated by several mechanisms, including hydrolyzed by proteases.
9. [3.1. IP-1 induced autophagy in mammalian cells] 1. The result 3.1 describes that IP-1 increased the number of HEK293T cells with more than 6 detectable autophagosomes per cell compared with cells not treated with IP-1 or an autophagy inducer (Trehalose), but Figure 1 seems that cells treated with Trehalose form autophagosomes than other cells treated with IP-1. Please describe in detail if it is not the error of sentence.
We appreciate the note. Indeed 100 mM trehalose induces more autophagosomes than 10 or 50 microM of IP-1. We have corrected this mistake. The amount of p62/SQSTM1 protein is reduced when it is properly degraded in autolysosomes, as a consequence of a functional autophagic flux. Considering the comments from another review, this figure was passed to supplementary material.
10. To explain the Figure 2B, the 3.1 paragraph says that an alternative method to monitor autophagic flux is by following the degradation of the autophagy receptor p62/SQSTM1, but the paragraph of introduction regarding p62/SQSTM1 states that the abundance of p62/SQSTM1 is indicative of a functional autophagic flux. I am confused about this conflict, so please add the details in the paragraphs.
We apologize for the confusion, which arises from an incorrect translation from Spanish to English language. Instead of “abundance”, it should say “amount”. We have corrected this problem in the text.
11. [3.2 Cytotoxicity of IP-1 at high dosages] 3. By explanation of Figure 4, I understood IP-1 at high dosages causes necrosis and apoptosis, but Figure 4 does not describe the measurement of IP-1 at 10uM although Figure 4 paragraph seems to focus on the result of IP-1 at 10uM and 50uM. Why?
We appreciate this note. The sentence where we concluded that “these results indicate IP-1 induced necrotic cell death on HEK293T cells and apoptosis with necrosis on MEF cells at concentrations higher than 10 μM.”, we meant to emphasize that the toxicity was observed at concentrations higher than the ones observed to induce autophagy (10 microM), but it is confusing. We have changed the writing to avoid this confusion.
12. [3.5 ATP quenching by IP-1] 4. To test hypothesis that small amounts of IP-1(10uM) may
induce autophagy, the experiment was conducted by monitoring the ATP quenching ability of IP-1. But in Figure 5, Intra and extracellular ATP concentration of cells exposed to IP- 1 at 10uM was not altered. How did autophagy happen at 10uM in spite of the constant ATP level?
This is an important question. The measured affinity of IP-1 for ATP (440 microM) indicates that IP-1 needs a large concentration of ATP to bind it; in agreement with this result, we observed that only the intracellular concentration of ATP was affected. Within cells, ATP concentration is maintained at 1-10 mM, while extra-cellularly is 10 nM (https://www.sciencedirect.com/science/article/pii/S0005273603002104). Thus, the intra-cellular ATP concentration is above the binding affinity for IP-1, thus IP-1 should be able to bind ATP. In contrast, the extra-cellular ATP concentration is not enough to bind to IP-1, even though this is where we expect most IP-1 would be. Since our measurements are global, we cannot establish at this point where inside the cells ATP concentration is suited for IP-1 to quench it. To clarify this, we have added a sentence in the Discussion section: “Our results also show that at 50 μM of IP-1, the ATP levels were more affected in the intracellular space than in the extracellular medium. This result is in agreement with the relatively large affinity-binding constant of IP-1 for ATP, since most ATP is inside cells. Yet, at 10 μM IP-1 induces autophagy and at such concentration no reduction of ATP is observed, suggesting that ATP at some particular location may be affected, but not globally; further experiments are required to test this idea.”.
Reviewer 2 Report
In this manuscript, authors described the antimycobacterial activity of the AMP IP-1. This peptide has intrinsic antibacterial activity and modulates macrophage properties to be negative for bacterial establishment. Among others, IP-1 peptide induces autophagy and cell dead. The research is well conducted and the conclusions are supporter by the results. However certain questions stay open and deserve more investigations. The manuscript can be considered to be accepted after revision.
The following suggestions need to be considered:
Major points:
- Authors demonstrated that IP-1 induces necrosis in HEK cells and apoptosis in MEF cells. Moreover, they observed cell death in macrophages. Authors should define if IP-1 induces necrosis or apoptosis in macrophages that are reservoir cells for Mtb infection. Depending of the results, authors should discuss it as these cell death process play paradoxical role during tuberculosis.
- Authors shown that IP-1 decrease intracellular ATP level leading autophagy and cell death. Moreover, I wonder if it could impact phagosomal acidification as this process is ATPase dependent. IP-1 could impair Mtb vacuolar acidification and modulate intracellular trafficking.
- Authors mentioned that IP-1 induces TNFa production in macrophages. However, if we look at the figure 7G, TNFa level is decreasing when IP-1 concentration increase…
- Figure 8: authors described intrinsic antibacterial activity of IP-1. What are the IC50 of the peptide for both strains? What is the mechanism of action? Could you prove that the peptide reaches the phagosome inside the macrophage to exert its antibiotic function?
- Authors use HEK cells to study cell dead and autophagy induction before to confirm it in macrophages (more accurate model to study TB). First results could be transferred to the supplementary file in order to decrease text length and facilitate the reading.
- For western blot analysis, authors used different loading controls between figure 1 (Tubulin) and Figure 2 (GAPDH). Could you use the same control for all experiments?
Minor points:
- Lines 327 and 337: put M. tuberculosis in italics
- Line 328: do you mean 3x10^5?
- Line 379: do you mean 2.5x10^5?
- Line 680: do you mean 5x10^6?
- Line 379: replace uL by µL
- Should be important to use the same units throughout the manuscript concerning IP-1 utilization (µg vs µM)
Author Response
Major points:
1. Authors demonstrated that IP-1 induces necrosis in HEK cells and apoptosis in MEF cells. Moreover, they observed cell death in macrophages. Authors should define if IP-1 induces necrosis or apoptosis in macrophages that are reservoir cells for Mtb infection.Depending of the results, authors should discuss it as these cell death process play paradoxical role during tuberculosis.
This is an important question, so we run a new experiment testing in chamber slides the morphology of infected or non-infected macrophages incubated with different concentration of IP-1. Infected macrophages incubated with 8μg/ml of IP-1 during five days show good cell preservation (55.8% survival cells) with many activated cells (very large with big nucleous), while incubation with 16 μg/ml showed many more death cells (95% of dead cells). It is well established that in TB, autophagy confers protection while necrosis disseminates the disease. Thus, it is mentioned in the discussion that it is important to administrate a low dose of IP-1 in order to induce autophagy, while high doses should not prevent the disease progression because it will induce necrosis.
2. Authors shown that IP-1 decrease intracellular ATP level leading autophagy and cell death. Moreover, I wonder if it could impact phagosomal acidification as this process is ATPase dependent. IP-1 could impair Mtb vacuolar acidification and modulate intracellular trafficking. Authors mentioned that IP-1 induces TNFa production in macrophages. However, if we look at the figure 7G, TNFa level is decreasing when IP-1 concentration increase…
Indeed, IP-1 could affect phagosomal acidification due to its activity to sequester ATP. We discard this possibility considering that our results show that IP-1 induces the full flux of autophagy; quenching ATP in the autolysosome may prevent the autophagy flux, hence we propose this ATP binding should take place somewhere else. The new results presented in figure S10 show that IP-1 is located at the autophagosomes (CytoID does not stain autolysosomes), where no ATP pump is present, hence the ATP quenching activity of IP-1 should not affect at this point the acidification. We have included this important aspect in the discussion section.
It is also now mentioned in the corresponding results section and figure legend that low concentrations of IP-1 induced high secretion of TNFα from macrophages, while high IP-1 concentration induced low TNFα probably because it is inducing cells death. This is in well concordance with the new experiment commented in the previous item. Thus, it seems that reduction of ATP induced by IP-1 is dose dependent and low or mild concentrations produced autophagy and TNFα production, while high concentrations induced necrosis.
3. Figure 8: authors described intrinsic antibacterial activity of IP-1.What are the IC50 of the peptide for both strains? What is the mechanism of action? Could you prove that the peptide reaches the phagosome inside the macrophage to exert its antibiotic function?
This is an important question, so we did a new experiment to determine the MIC and IC50, which is 92.66 μM (230.78 μg/ml) for MDR strain and 99.27 μM (247.25 μg/ml) for H37Rv strain. We suppose that the same quenching ATP mechanism to induce cell death by IP-1 is also producing bacterial killing. We used an IP-1 peptide conjugated with TAMRA, a red fluorophore, which allowed us to observe its intracellular localization. To determine whether the peptide could reach intracellular MBT, we simultaneously detected autophagosomes with CytoID, a specific reagent that fluoresces in green. As shown in figure S10, indeed, some autophagosomes could be detected with IP-1 peptide inside.
4. Authors use HEK cells to study cell dead and autophagy induction before to confirm it in macrophages (more accurate model to studyTB). First results could be transferred to the supplementary file in order to decrease text length and facilitate the reading.
We have moved to supplementary data the first 2 figures referring to autophagy induced by IP-1 in HEK and MEF cells. Some of the explanatory text related to these figures was moved as well to supplementary material to effectively reduce the length of the main document.
5. For western blot analysis, authors used different loading controls between figure 1 (Tubulin) and Figure 2 (GAPDH). Could you use the same control for all experiments?
The proteins used as a loading reference were chosen according to the molecular weight of the protein to be analyzed.
Minor points:
Lines 327 and 337: put M. tuberculosis in italics
Line 328: do you mean 3x10^5?
Line 379: do you mean 2.5x10^5?
Line 680: do you mean 5x10^6?
Line 379: replace uL by μL
Should be important to use the same units throughout the manuscript concerning IP-1 utilization (μg vs μM)
We appreciate these notes. We have fixed them in the new version of our work. Regarding the units, we prefer to keep the amount of peptide used to treat animals in μg per 100 μL because the peptide when it is acting on the lungs of the animals gets diluted and hence it is not at the specified concentration.
Round 2
Reviewer 1 Report
My concerns have been addressed.
pharmaceutics-940839 was accepted!